# Polycomb Recruiters Inside and Outside of the Repressed Domains

**DOI:** 10.3390/ijms241411394

**Published:** 2023-07-13

**Authors:** Maksim Erokhin, Vladic Mogila, Dmitry Lomaev, Darya Chetverina

**Affiliations:** Institute of Gene Biology, Russian Academy of Sciences, 34/5 Vavilov Street, Moscow 119334, Russia; v@mogila.org (V.M.); lomaevdv@gmail.com (D.L.)

**Keywords:** Polycomb, Trithorax, Mediator, TFIID, NELF, SWI/SNF, H3K27me3, PRE, enhancer

## Abstract

The establishment and stable inheritance of individual patterns of gene expression in different cell types are required for the development of multicellular organisms. The important epigenetic regulators are the Polycomb group (PcG) and Trithorax group (TrxG) proteins, which control the silenced and active states of genes, respectively. In *Drosophila*, the PcG/TrxG group proteins are recruited to the DNA regulatory sequences termed the Polycomb response elements (PREs). The PREs are composed of the binding sites for different DNA-binding proteins, the so-called PcG recruiters. Currently, the role of the PcG recruiters in the targeting of the PcG proteins to PREs is well documented. However, there are examples where the PcG recruiters are also implicated in the active transcription and in the TrxG function. In addition, there is increasing evidence that the genome-wide PcG recruiters interact with the chromatin outside of the PREs and overlap with the proteins of differing regulatory classes. Recent studies of the interactomes of the PcG recruiters significantly expanded our understanding that they have numerous interactors besides the PcG proteins and that their functions extend beyond the regulation of the PRE repressive activity. Here, we summarize current data about the functions of the PcG recruiters.

## 1. Introduction

During development of the multicellular organisms, individual patterns of gene expression are established in each cell type and stably transmitted through many consecutive cell divisions [1]. The precise control of gene expression in each tissue involves an antagonism between the Polycomb transcriptional repressors (PcG) and the Trithorax activator proteins (TrxG), which regulate the activity of the facultative heterochromatin and the developmentally regulated euchromatin, respectively [2,3,4,5].

The antagonism between the PcG and TrxG proteins was first demonstrated in *Drosophila* through studies on the Hox genes that specify the correct segmentation pattern of the body [3,5,6,7]. Mutations in the PcG-encoding genes lead to characteristic homeotic transformations due to an overexpression of the Hox genes, while mutations in the TrxG-encoding genes dominantly suppress these phenotypes. The loss-of-function mutations in the TrxG-encoding genes can also cause homeotic transformations, but they instead occur due to insufficient expression of the Hox genes [3,5,6,7]. The subsequent studies have shown that, besides the Hox genes, the PcG/TrxG proteins control transcription of many developmental genes involved in different cellular processes and dysfunctions of genes encoding the PcG/TrxG proteins lead not only to various developmental abnormalities but to cancer as well [8,9,10,11,12,13].

In *Drosophila*, the PcG/TrxG group proteins are recruited to the specialized DNA regulatory sequences—the Polycomb response elements (PREs) [6,14,15,16]. It was shown that PREs act as silencers and maintain repression of reporter genes throughout development in transgenes, thus constituting independent subunits of repression. However, at least some PREs demonstrate switchable dual properties and, under certain conditions, can activate transcription [6,14,15,16,17]. In addition, a number of developmentally regulated embryonic enhancers possess PRE activity in adults [18], indicating that at least some PREs in the activating state can represent classical enhancer elements. In accordance with the dual activity of some of the PREs, they can recruit both the PcG and TrxG proteins. It was suggested that the resulting activity of the PREs is the outcome of competition between the PcG and TrxG proteins [6,14,15,16].

A number of the PRE DNA-binding proteins, named PcG recruiters, which are implicated in the PRE repressive activity, have been identified [14,15,16]. However, genome-wide studies have shown that the binding of the PcG recruiters, as well as PcG core proteins, is not restricted to the H3K27me3 domains [19,20,21,22,23,24,25]. In particular, they can be recruited to the promoters of active genes and to the enhancers [20,23,24] and in a number of cases they were shown to be implicated in their activities. In accordance with these studies, recent analysis of the interactomes of the PcG recruiters demonstrate that each of them has a unique set of protein partners, which in addition to the PcG repressors, include components of the Trithorax complexes, as well as proteins of the basic transcriptional machinery, e.g., constituents of the promoter pausing, the Mediator complex, and the architectural proteins.

In this review, we will summarize the known data on the participation of the PcG recruiters in the PRE repressive functions, as well as outside of the H3K27me3 domains, and will focus on the partners of the PRE DNA-binding proteins implicated in distinct regulatory activities.

## 2. PcG and TrxG Proteins

The majority of the known PcG proteins are organized into multisubunit complexes which are targeted to the chromatin and which mediate the repression of transcription [4,26]. The main PcG complexes include the Polycomb repressive complex 2 (PRC2) and the Polycomb repressive complex 1 (PRC1) (Figure 1).

The PRC2 contains the Enhancer of zeste (E(z)), Suppressor of zeste 12 (Su(z)12), Extra sex combs (Esc), and the Chromatin assembly factor 1 (Caf1) subunits [27,28]. The SET domain of the E(z) protein possesses a methyltransferase activity and catalyzes the H3K27me3 histone modification [27,28] that is specific for the chromatin regions repressed by the PcG proteins [29,30].

The Polycomb repressive complex 1 (PRC1) contains Polycomb (Pc), Polyhomeotic (Ph), Sex combs extra (Sce, also known as dRing), and Posterior sex combs (Psc) subunits [31,32,33]. The Sce is an E3 ubiquitin-protein ligase that catalyzes the H2AK118ub modification [34]. Polyhomeotic (Ph) is encoded by two paralogous genes: *ph-p* and *ph-d,* giving rise to almost identical proteins. The Psc protein can be replaced in the complex by its homolog, the Su(z)2 protein [35]. The core of PRC1 can compact chromatin, inhibit nucleosome remodeling, and repress transcription [31,33,36,37,38]; the main role in these processes belongs to Psc and Su(z)2 [35,36,37,38]. The Pc protein of the PRC1 interacts with the H3K27me3 histone modification via chromodomain [39,40].

In addition to the core subunits, a number of minor proteins could be co-purified with the PRC1 and PRC2. One such PcG protein that co-purifies with both the PRC1 and PRC2 is the Sex comb on midleg (Scm) [32,33,41]. Scm can directly interact with Ph [42,43] but is recruited to chromatin independently of the PRC1 or PRC2 [44,45].

The TrxG proteins include subunits of different complexes involved in the activation of transcription (Figure 1). In particular, the TrxG proteins include the ATP-dependent remodelers of the SWI/SNF family complexes, Trx, Trr, and CBP proteins [2,5,46].

*Drosophila* has two complexes that belong to the SWI/SNF remodeler family—PBAP and BAP. The catalytic subunit of both complexes is the Brahma (Brm) ATPase [47,48]. Using ATP, chromatin remodelers change the structure, assembly, and arrangement of histones on the DNA, contributing to the decompactization of chromatin and to the recruitment of the activator complexes [49]. In addition to Brm, PBAP and BAP have six common subunits—Bap55, Mor, Snr1, Bap60, Bap111, and Act5C. The unique subunits of the PBAP complex are Polybromo (PB), Bap170, and SAYP, while the BAP complex has Osa [47,48].

The Trithorax (Trx) and Trithorax-related (Trr) proteins are histone methyltransferases that catalyze the H3K4me1/2 modification characteristic for the active chromatin regions [50,51,52] and they are subunits of the COMPASS-like complexes [2,53].

The CBP protein, encoded by the *nejire* gene, is an acetyltransferase that performs histone modifications at a subset of histone positions, including the H3K27ac modification, a hallmark of the active chromatin [54]. Histone acetylation reduces the strength of the nucleosome–DNA interaction and leads to a decompactization of the chromatin [55].

In *Drosophila*, Trx, Trr, and CBP, as well as the H3K4me1/2 and H3K27ac modifications, are co-localized on the active enhancers and at the regions of PcG protein recruitment outside of the H3K27me3 domains [50,51,52,54].

## 3. PcG Recruiters

PREs can be located at different distances from the target genes, both in the immediate vicinity of the transcription start site and at a distance of thousands of base pairs. Structurally, they are DNA fragments of several hundred base pairs in length. Like other regulatory elements, they are DNA regions with a reduced density of nucleosomes and are hypersensitive to DNase I. PREs contain binding sites for the PcG recruiters. The sites for the PcG recruiters in each particular PRE can be present in different combinations, in different numbers and positions relative to each other.

The currently characterized PcG recruiters are Pleiohomeotic (Pho) [56,57] and its close homolog Pleiohomeotic-like (Phol) [58], Combgap (Cg) [59], Sp1 factor for pairing-sensitive silencing (Spps) [60], Crooked legs (Crol) [61], GAGA-factor (GAF, also known as Trl) [62,63], Pipsqueak (Psq) [64,65], Zeste (Z) [32,66], Dorsal switch protein 1 (Dsp1) [67], Grainyhead (Grh, also known as NTF-1) [68], and Alcohol dehydrogenase transcription factor 1 (Adf1, also known as Nalyot) [69] (Figure 2).

Six of the eleven enumerated proteins, Pho, Phol, Combgap, Spps, Crol, and GAF, belong to the class of the C2H2-type Zinc finger (C2H2-type ZnFs) proteins (Figure 2). Pho and Phol have four C2H2-type ZnF motifs, which are 80% identical and interact with the sequences having a GCCAT core in PREs [57,58,70] and are genome-wide [71,72,73,74]. Combgap contains 11 C2H2-type ZnFs motifs and interacts with (GT)n [59]. Spps has three C2H2-type ZnF motifs and belongs to the Sp1/KLF family. Like all the members of this family, Spps interacts with the RRGGYG sequences (R = A or G, Y = C or T) [75]. Crol contains 18 C2H2-type ZnF motifs and binds to poly(G)-rich sequences [61]. GAF contains a single ZnF at the central part of the protein that binds to the (GA)n [76]. In addition, the N-terminus of GAF has a conserved Bric a brac, Tramtrack, Broad-complex/Poxvirus, Zinc finger (BTB/POZ) domain, which is involved in self-oligomerization and heterologous protein–protein interactions.

Each of the other five PcG recruiters has DNA-binding domains of distinct types (Figure 2). Psq contains four HTH DNA-binding domains at the C-terminus. While the Psq DNA-binding domain type is different from that of GAF, it was also shown to recognize the (GA)n sequences [19,77,78]. Another similarity with GAF is the presence of the BTB/POZ domain at the N-terminus of Psq.

Zeste contains the Myb/SANT-like DNA-binding domain that binds YGAGYG [19,79,80].

Dsp1 contains two High mobility group (HMG) motifs. Initially, Dsp1 was reported to bind to the GAAAA sequence in the *Fab7*PRE in transgenic constructs [67]. However, the genome-wide Dsp1 binding site was identified as (GA)n [74]. It has been reported that the HMG proteins are able to bind DNA with no sequence specificity [81,82,83], suggesting that Dsp1 can mimic the overall Ph peak consensus, which was shown to be (GA)n as well. Interestingly, the most enriched ChIP-seq motifs for Spps and Zeste were also identified as (GA)n rich [19,20], indicating that the regions of their binding highly overlap with the (GA)n-rich sequences.

Grainy head (Grh) contains the Grh/CP2 DNA-binding domain that was first shown to bind to YNAACYGGTYYTGCGG in vitro [84]; a similar AACYNGTTT core was identified by a genome-wide study [85]. Adh transcription factor 1 (Adf1) contains an MADF DNA-binding domain at the N-terminus that binds ACGBCGRC (B = T, C, or G) [19]. In addition, Adf1 has a C-terminal BESS domain involved in self-dimerization and protein–protein interactions [86].

## 4. Role of PcG Recruiters in PcG Repression

The *pho* mutants die at the pharate adult stage and have sex combs on the second and third legs, demonstrating a classic homeotic PcG phenotype [87,88]. Accordingly, mutations of the *pho* gene lead to a derepression of the Hox genes [57,58,89]. *pho* mutations affect the ability of the transgenic PRE to repress reporter genes and so do mutations of its binding sites in the PRE [3,15,16]. However, Pho sites alone fail to make up a functional PRE [70,90] and, thus, require other proteins. For example, *phol* mutation does not have homeotic phenotype, but it enhances the *pho* mutation [58]. In addition, while the recruitment of the Pc (PRC1), E(z), and Su(z)12 (PRC2) to the *bxd*PRE is sensitive to the Pho knockdown by RNAi in *Drosophila* cells where Phol is absent, the double *pho/phol* mutations are required for the loss of Pc binding at the *bxd*PRE at the larval stage where both proteins are expressed [91]. Accordingly, both the Pho and Phol proteins are required for the binding of Pc, Psc, Scm, and E(z) to several sites on the larval polytene chromosomes [58].

The *dsp1* mutants have a mixed PcG/TrxG phenotype, indicating that Dsp1 is most likely involved in both repression and activation of transcription [92]. Mutations of other PcG recruiters do not have a PcG homeotic phenotype, but the majority of them were shown to enhance mutations in the *PcG* core genes or *pho*, leading to more severe homeotic phenotypes. For example, the *Trl* [63,64,70], *Psq* [64,65], and *Adf1* [69] mutations enhance *PcG* mutants, and the *grh* [68] mutation enhances *PcG* and *pho* mutants. The *cg* (encoding Combgap) [59] and *Spps* [60] mutations were shown not to affect the homeotic phenotype of *pho* mutants but enhance lethality.

Combgap, Spps, Crol, GAF, Psq, Zeste, Dsp1, and Grh were confirmed to be important for the functioning of transgenic PREs either by genetic tests and/or by mutations of their binding sites ([61] and reviewed in [15]). For example, *crol* knockout and mutation of the Crol binding site decrease the silencing caused by transgenic *eve*PRE and affect recruitment of Ph and Combgap [61]. *Trl* mutations decrease the silencing caused by the *Fab7*PRE [62]. Dsp1 binding to *Fab7*PRE was shown to be sensitive to *dsp1* mutation on larval polytene chromosomes, and the Dsp1 G(A)n site affected recruitment of PRC1 (Ph) and PRC2 (E(z)) [67].

## 5. Interactions between PcG Recruiters and PcG Core Complexes

In immunoprecipitation experiments, Pho was shown to interact with the Scm-related gene containing four malignant brain tumor (MBT) domains protein (Sfmbt) [93] (Figure 3). Structural analysis revealed that the spacer region of Pho forms a tight complex with the four MBT domains of Sfmbt [93]. Sfmbt is a PcG member, since mutations in the *Sfmbt* gene lead to misexpression of the Hox genes [94]. Since both Pho and Sfmbt are recruited to PREs, this protein pair was assigned to a separate complex named Pho repressive complex (PhoRC). Both Pho and Sfmbt have been shown to have a number of physical contacts with the PcG proteins (Figure 3). Pho co-immunoprecipitates with the components of the PRC1 and PRC2 [95,96,97] and interacts directly with Ph and Pc [98] (PRC1) and E(z) and Esc [91] (PRC2). In turn, Sfmbt interacts with the Scm protein in the direct and indirect assays [95,99].

In addition to Pho, the Phol protein also interacts with Sfmbt. However, it was shown that Pho and Phol interact with Sfmbt in a mutually exclusive manner [94]. A quantitative analysis of the Phol complex in vivo has not yet been carried out, but Phol has been shown to interact with the Esc subunit of PRC2 in a pull-down assay [91].

For Spps, contacts with the PRC1 and PRC2 core subunits have not been described, but it is able to directly interact with the Scm protein [100].

Of the other PcG recruiters, Combgap [59], Crol [61], GAF [96], Psq [101], Zeste [32], Grh [97], and Adf1 [69] were co-purified with the PRC1 subunits. Of these, Zeste and Grh interact directly with Ph [102] and Sce [103], respectively. In addition to PRC1, GAF can be co-purified with PRC2 [96,104], but no direct contacts with the PRC1 and PRC2 have been found for GAF. Our recent IP/LC–MS analysis showed that, at least in S2 cells, Combgap and Zeste show a stronger association with PRC1 than Psq and Adf1 [19].

## 6. Connections between PcG Recruiters and TrxG Proteins

Several observations indicate that Dsp1, GAF, Zeste, and Pho/Phol PcG recruiters have TrxG-related activities. As mentioned above, *dsp1* mutants have a mixed PcG/TrxG phenotype [92] and, in particular, resemble and enhance the *Ubx* loss-of-function phenotype. Enhancement of the *Ubx* phenotype is also observed in genetic experiments with the *Trl* gene. In addition, hypomorphic *Trl* alleles have loss-of-function transformations in segment A6 [105]. Zeste was shown to be necessary for the PRE-mediated inheritance of the active chromatin state of a small *Fab7*PRE fragment [106] and plays an activating role in the case of the *bxd*PRE element [107]. In the case of the *Fab7*PRE, Zeste sites were required for recruitment of the SWI/SNF TrxG Brm protein to the transgene [106]. In addition, Pho and Phol are required for the maintenance of both the repressed and active state of *eve*PRE [108].

A number of PcG recruiters were shown to interact with the TrxG SWI/SNF ATP-chromatin remodelers. GAF was shown to co-purify both with BAP and PBAP [104,109]. Pho [98], Psq [19], and, less efficiently, Combgap [19] and Zeste [110] were shown to precipitate with the BAP complex (Figure 3). In our IP/LP-MS experiments, Psq interacted with the BAP proteins much more efficiently than with the PcG proteins, that were present only at background levels [19]. Also, Pho was shown to directly bind to Brm [98], and Zeste with Mor and Osa [110].

In addition, PcG recruiters were shown to co-purify with several other TrxG proteins [19,111]. In particular, Zeste and Psq efficiently precipitated the Fs(1)h TrxG protein [19], which was previously shown to co-purify with PRC1 and to be correlated with Pc and the H3K27ac modification genome-wide [21]. Fs(1)h also co-purified Pho and GAF proteins [21]. Importantly, Fs(1)h recognizes the same site as Zeste and, thus, has a potential to be directly bound to the PRE elements.

## 7. PcG Recruiters Are Localized Outside of H3K27me3 Domains Genome-Wide

The functional and physical connections of the PcG recruiters with the TrxG proteins indicate their importance for the PRE activating state and, potentially, a more global role in the activation of transcription. Accordingly, current data suggest that the binding of PcG core proteins, as well as PcG recruiters, is not restricted to the H3K27me3 domains [19,20,21,22,23,24,61]. There are also indications that PcG recruiters are important for the PcG recruitment outside of the H3K27me3 domains. In particular, it was confirmed for Spps, Pho [20], Combgap [20,59], and Crol [61] recruiters with respect to the Ph protein.

If we evaluate the level of overlap of the PcG recruiters with the PcG proteins, then about 80% of the Pho, Combgap [59], and Crol peaks [61] will overlap with Ph at the 3rd instar larval imaginal discs and brains and this will constitute over 60% of the Ph sites (Figure 4A). In *Drosophila* S2 cells, approximately 80% of Combgap, Psq, and Zeste overlap with Ph. In the case of Adf1, the overlap with Ph is about 50% [19] (Figure 4A).

However, if we consider the overlap relative to the PREs (PcG core proteins and the H3K27me3 histone modification), then the intersection for Pho [61], Combgap [19,61], Spps, Crol [61], Psq, Zeste, and Adf1 [19] will be about 7 to 20% (Figure 4B).

Accordingly, over 80% of peaks for PRC1 (Ph, Psc) and PRC2 (E(z)) fall outside of the H3K27me3 domains as well [20] (Figure 4C). For Pc, the percent of peaks outside of H3K27me3 was estimated to be about 65% [20] (Figure 4C). The average Ph [23] and E(z) [20] binding levels are lower outside of the H3K27me3 domains. The E(z) was suggested to be inactive due to the presence of active chromatin marks [20]. However, additional studies are required to understand what can block E(z) H3K27me enzymatic activity.

## 8. PcG Recruiters at Active Promoters and Enhancers

Going back in history, GAF [112,113,114], Grh [84,115], Zeste [79,116], and Adf1 [117] were initially discovered as the promoter-bound activators of transcription. In particular, GAF, Grh, and Zeste together positively regulate the transcription of the *Ubx* gene [84,118,119]. Subsequent studies have shown that all currently known PcG recruiters can bind to the actively transcribed promoters [20,61,69,71,74,85,120]. A number of the PcG recruiters were confirmed to bind to the predicted enhancers (Combgap, Zeste, Psq, Adf1, GAF, and Grh) [19,85,121]. Grh [85] and Zeste [122,123] were confirmed to be functionally required for enhancer activity in a number of cases. Likewise, PRC1 has also been shown to be recruited to the promoters of active genes and to potential enhancers [23,24].

For several PcG recruiters, the estimated overlap was quantified. GAF binds to 20% of active promoters [120] and binds to 53% of HOT elements predicted to function as enhancers [121]. In the S2 cells, in the range of 19 to 41%, Combgap, Zeste, Psq, and Adf1 overlap with the promoters having H3K4me3 and, thus, are suggested to be active [19]. In addition, 8 to 13% of Combgap, Zeste, Psq, and Adf1 overlap with potential active enhancer regions [19] defined by the H3K27ac peaks that are bound by CBP and localized outside of TSS. A high degree of overlap with the predicted active enhancers is shown for Psq in Kc167 cells, where it was shown to overlap with the active histone modifications, CBP, and also with GAF and Pc [78].

In agreement with the functional overlap of the PcG recruiters with enhancers and promoters, the interactomes of the PcG recruiters contain many proteins implicated in the activities of these regulatory elements.

### 8.1. PcG Recruiters and Their Promoter Partners

The Pol II together with the general transcription factors (GTFs: TFIIA, TFIIB, TFIID (composed of TBP and more than 10 TAFs), TFIIE, TFIIF, and TFIIH) are recruited to the promoters and they are assembled into the preinitiation complex (PIC). A successful formation of the PIC leads to a rapid initiation of transcription. Pol II pauses after it produces an RNA of 20–60 nucleotides in length. Current data suggest that the pausing of Pol II at the early elongation is an obligate part of the transcription cycle; a step that is universally encountered by Pol II for the majority of even highly expressed genes. The promoter pausing is primarily mediated by an action of the NELF and DSIF complexes that stabilize Pol II at the pause region [124,125]. In addition, the M1BP DNA-binding factor, the RNA polymerase II-associated factor 1 complex (PAF1C), and TFIID were shown to be implicated in the promoter pausing [126].

Of the PcG recruiters, GAF is well known to be implicated in pausing for a large subset of genes, most of which are developmentally regulated [111,120,127,128,129,130,131]. In total, GAF associates with about 20% of the Pol II-bound promoters and correlates with the highest levels of paused Pol II [120,128]. Of the promoter-associated proteins, GAF was shown to interact directly and in co-IP with Taf3 [132,133] and Taf4 [104,133] subunits of the TFIID complex [132] (Figure 3).

In a pull-down assay, Grh interacted with Taf2 and Taf6 TFIID subunits [134] (Figure 3). In IP/LC-MS, Combgap efficiently co-purified with a number of the TFIID subunits (Figure 3), including Taf4, Taf7, Taf8, and Taf9 [19]. In addition, Combgap shows stable association with the NELF complex (Figure 3) and M1BP. We have recently shown that over 80% of Combgap overlaps with the NELF-A and NELF-B subunits, genome-wide. Moreover, Combgap is required for an effective NELF recruitment to two genes, *CG4562* and *Pp2B-14D*, which are actively transcribed, as well as to the *Poxm* gene, which is under the H3K27me3 repression [19].

Pho can directly interact with the Spt5 subunit of DSIF [135,136] (Figure 3) and shows about 70% overlap with Spt5 and NELF, genome-wide [136].

It was shown that, except for promoters, NELF [19,137] and DSIF (Spt5) [23,138] are bound to the PREs/enhancers. Moreover, the *spt5* and *nelf-A* mutants enhance the PcG phenotype observed in *pho* mutant flies, suggesting an involvement of pausing factors in the PcG silencing [136]. Thus, both the PcG recruiters and the pausing proteins have a potential to be co-regulators of promoters and PREs/enhancers.

Not surprisingly, PRC1 is connected to promoter functioning as well. It was demonstrated that depletion of the PRC1 subunits by RNAi alters both phosphorylation of RNAP II and recruitment of the Spt5 subunit of the DSIF complex to the active genes [23] and decreases gene transcription [24].

### 8.2. PcG Recruiters and Mediator Complex

Mediator is a large co-activator complex consisting of about 30 subunits, named MED1–MED31, Cdk8, and cyclin C [139]. Mediator associates with enhancers and active promoters and a depletion of Mediator subunits reduces transcription of the enhancer-controlled genes [140,141,142,143].

Adf1 stands apart from other PcG recruiters in the way that it highly efficiently co-purifies with the Mediator complex (Figure 3). Moreover, over 55% of the Adf1 peaks overlap with the MED1 and MED30 Mediator subunits genome-wide [19]. This suggests that Adf1 might be involved in the recruitment of Mediator to the chromatin. Intriguingly, while not as efficient as Adf1, Fs(1)h [21] and, less so, Psq [19] were also shown to co-purify a number of Mediator subunits.

### 8.3. PcG Recruiters and Architectural Proteins

The architectural (also known as insulator or boundary) proteins interact with the regulatory elements—boundaries that can limit enhancer–promoter interactions and block the spread of the heterochromatin [144,145]. They are also required for the separation of the genomic domains into TADs and for the formation of the 3D chromatin structure [146].

Of the PcG recruiters, GAF was experimentally confirmed to be required for the boundary activity of particular insulators [111,144]. GAF can co-purify with a number of architectural proteins (Figure 3), including Mod(mdg4)2.2, CP190, Chro, and Su(Hw) [104]; of these, GAF directly interacts with Mod(mdg4)2.2 [147,148,149]. However, genome-wide it does not show an extended overlap with the architectural proteins [150], suggesting that it might function differently from the other architectural proteins. Accordingly, GAF was shown to be implicated in the activity of tethering elements required for proper enhancer–promoter and promoter–promoter interactions of the developmentally regulated genes [151,152,153].

It was shown that 22–54% of the Combgap, Zeste, Psq, and Adf1 genome peaks overlap with the architectural protein CP190 in S2 cells [19]. In addition, for Psq, a high degree of overlap with CP190, Mod(mdg4)2.2, and Su(Hw) was shown in Kc167 cells [78]. In regard to physical interactions, Combgap was shown to precipitate different architectural proteins, including CP190, Chro, Mod(mdg4), Ibf1, Su(Hw), BEAF-32, and Ibf2 [19]. Psq can precipitate Ibf1, BEAF-32, Clamp [19], and Mod(mdg4) [78]. Adf1 purifies Mod(mdg4), Ibf1, BEAF-32, and Clamp [19]. Zeste failed to efficiently precipitate the boundary proteins in our recent study, but was previously shown to precipitate with Su(Hw) and Mod(mdg4) and to interact directly with Mod(mdg4) [154] (Figure 3).

While there is a huge overlap of Combgap, Zeste, Psq, and Adf1 with the boundary proteins genome-wide, it should be noted that CP190 frequently overlaps with promoters [150,155], and, moreover, boundaries are frequently found in close proximity to PREs [62,156,157,158] and can stimulate the PRE repressive activity [159]. Thus, the genome-wide functional consequences of the overlap with CP190 require more studies. In a recent study, Combgap sites were shown to be enriched at the Su(Hw)-indirect peaks [160], suggesting that, at least in some cases, the overlap can be a result of a non-direct recruitment to the regulatory elements. It will be important to estimate in the future the intensity of the peaks that overlap between the PcG recruiters and boundaries.

## 9. Interactions between PcG Recruiters and Model of Combinatorial Recruitment

The study of the interactomes of the PcG recruiters revealed that they have many contacts with each other. The strongest of the known contacts in the co-IP experiments are observed between Crol and Combgap/Pho [61]; between Combgap and Adf1/GAF; between Zeste and Psq/Dsp1 [19]; between Psq and GAF [19,104,111] (Figure 3). Moreover, direct interactions have been shown for the following pairs: Pho directly interacts with Spps [19], GAF [100], and Grh [68]; Combgap with Crol [61] and Adf1 [19]; GAF with Psq [147,161]; Psq with Dsp1 [19]. Except for the heterologous interactions, Pho, Spps, Combgap [19], GAF [147,162], Psq [147], Zeste [19,163], Grh [115], and Adf1 [19,164] displayed self-interactions. This suggests that they can display a higher affinity for the DNA fragments containing several closely placed binding sites that they recognize.

In accordance with the interactions between the PcG recruiters, for some of them, Pho, Combgap, Spps, Crol, Psq, Zeste, and Adf1, a significant genome-wide overlap was reported [19,20,59,61]. At the 3rd instar larval imaginal discs and brains, about 80% of Pho peaks co-localize with Combgap [20,59] and Crol [61], and this corresponds to ~50% of Combgap and Crol peaks (Figure 5). Crol displays 80% overlap with Combgap [61]. In addition, over 70% of the Spps sites overlap with Combgap, Pho, and Crol, and this corresponds to ~20–40% of Combgap, Pho, or Crol peaks [20,61]. In *Drosophila* S2 cells, 88% of Psq peaks overlap with Combgap, while the majority of Zeste peaks overlapped with Combgap (90%) and Psq (88%) [19]. The overlap of PcG recruiter with Adf1 is lower, but still exceeds 50% in the case of Combgap and Zeste sites. It should be noted that, overall, the binding profiles of the PRE DNA-binding proteins are unique and each of the tested proteins has its own unique binding peaks [19,20,59] and if we calculate simultaneous overlap between a larger number of recruiters together, they would show a lower frequency. For example, simultaneous overlap for Combgap, Zeste, Psq, and Adf1 proteins identifies 1168 regions which correspond to 15% of Combgap, 24% of Adf1, 27% of Psq, and 56% of Zeste peaks [19].

In addition, it should be noted that the extent of overlap can be different within and outside of the H3K27me3 domains. For Spps–Pho, it was reported to be lower outside of the H3K27me3 domains—about 90% and 70% of Spps peaks overlap with Pho, inside and outside of the H3K27me3 domains, respectively [20]. Another study suggested a better overlap of Grh with GAF at Grh-repressed loci [85]. In any case, the total extent of the mutual overlap between the PcG recruiters is substantially high, which allows the proposal that at many loci inside and outside of the H3K27me3 domains they can function together.

Several observations support a co-operative activity of the PcG recruiters. It was shown that Pho protein binding to a chromatinized PRE in vitro requires the assistance of GAF or Zeste proteins [165]. In another in vitro study, Pho and Grh facilitated each other’s binding to the naked DNA [68]. In vivo, Spps was shown to be important for the Pho recruitment at subsets of sites genome-wide, independent of the H3K27me3 state [20]. Similar, Crol is important for Combgap binding at several sites in and out of H3K27me3 domains [61].

In addition to the co-operation, the activities of the PcG recruiters were proposed to be redundant and partially compensate for each other [15,20].

In connection with the potential network that can be formed by multiple interactions between the PRE DNA-binding proteins, their functional co-operation, and redundancy, we have proposed a model according to which the combination of the PRE DNA-binding factors at the PRE forms a “platform” that provides for an efficient recruitment of the core PcG/TrxG complexes [15]. We propose that this model can be relevant not only at the PREs, but also outside of the H3K27me3 domains as well.

## 10. Conclusions

The current data suggest that the PcG recruiters, as well as PcG core proteins, bind to regions inside and outside of the H3K27me3 domains and participate in both the repression and activation of genes. Although much progress has been made toward an understanding of the role of the PcG recruiters in assembly of the PcG protein complexes genome-wide, many questions remain to be elucidated.

Multiple interactions between the PcG recruiters were identified and they show an extensive overlap with each other and with the PRC1 genome-wide. However, the currently described interactomes of the PcG recruiters, GAF, Combgap, Zeste, Psq, and Adf1, differ substantially from each other. This might suggest that each protein can attract a unique set of transcription factors to the DNA, and the presence of a binding site for a particular DNA-binding protein could subtly modulate the resulting activity of the DNA regulatory element. This can probably apply to both PREs and the other types of regulatory elements. The characterization of the interactomes of other PcG recruiters may provide a better understanding of the role of each of them in the genome-wide transcriptional regulation.

## Figures and Tables

**Figure 1 ijms-24-11394-f001:**
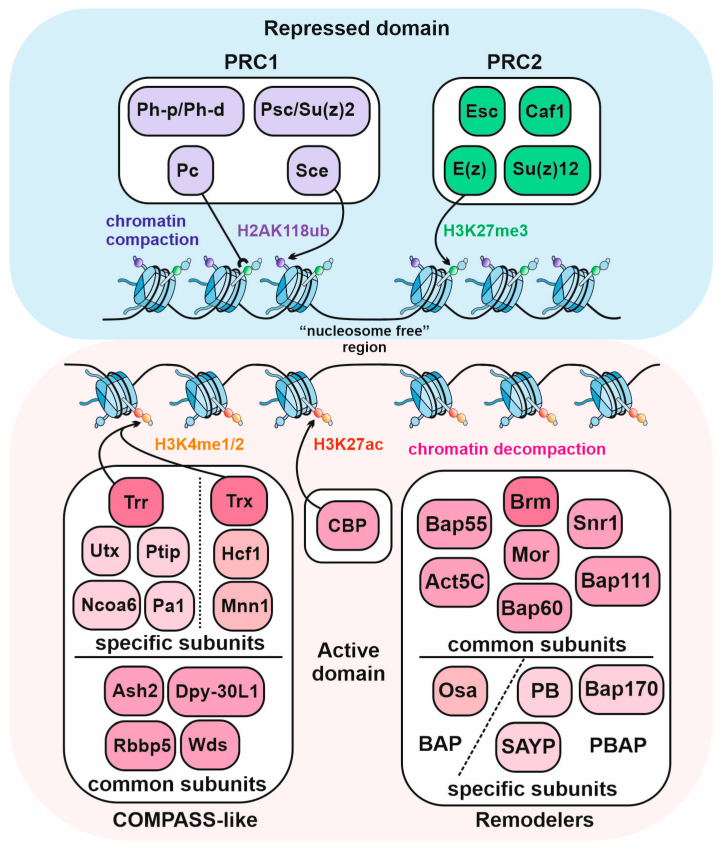
Functional activities of the *Drosophila* PcG and TrxG proteins. The E(z) subunit of the PRC2 complex catalyzes the H3K27me3-specific histone modifications in the PcG repressed domains. The PRC1 compacts the chromatin and its subunit Sce creates the H2AK118 ubiquitin modification. The Pc subunit of the PRC1 can interact with the H3K27me3 chromatin modification. The TrxG proteins, Trr and Trx, catalyze the H3K4me1/2 histone modifications and are subunits of the COMPASS-like complexes. The CBP is an acetyltransferase and it performs the H3K27ac histone modification. The BAP and PBAP complexes are the ATP-dependent chromatin remodelers belonging to the SWI/SNF family. The H3K27ac modification and SWI/SNF decompact the chromatin.

**Figure 2 ijms-24-11394-f002:**
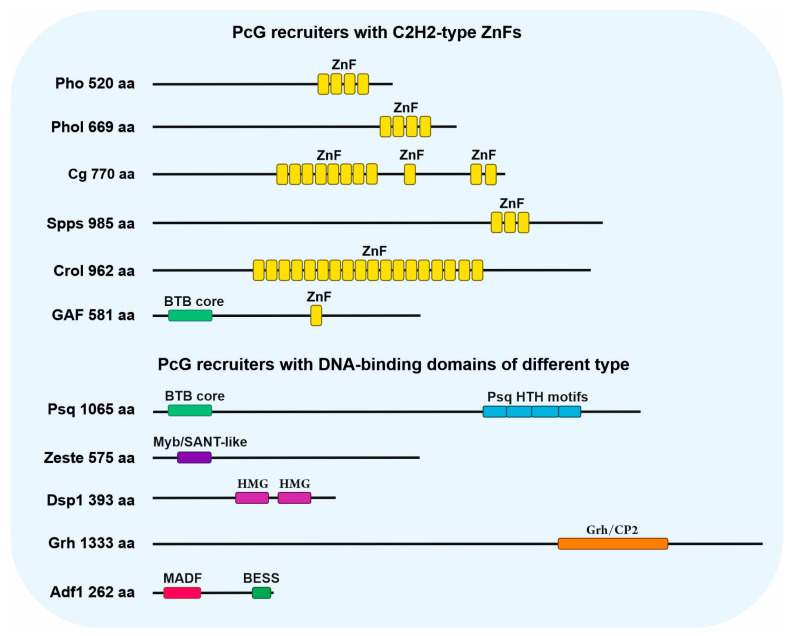
Structure of the PcG recruiter proteins. The C2H2-type Zinc finger DNA-binding motifs (ZnF) are shown in yellow, the Psq HTH motifs in blue, Myb/SANT-like in purple, HMG in magenta, Grh/CP2 in orange, MADF in red. The BTB and BESS protein–protein interacting domains are shown in green.

**Figure 3 ijms-24-11394-f003:**
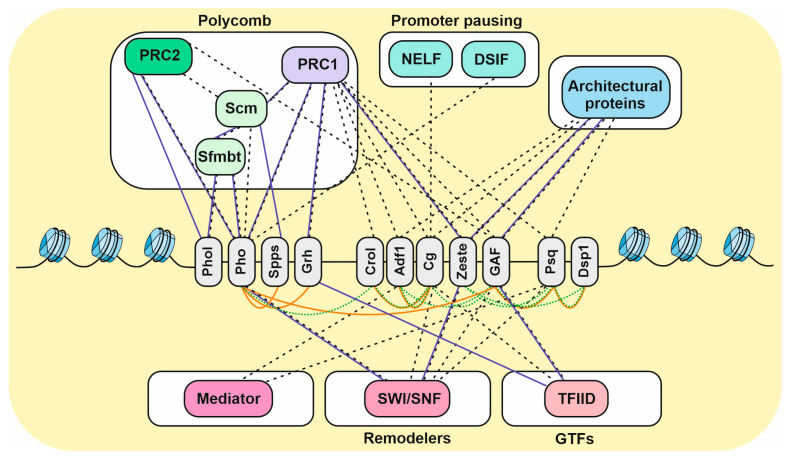
The physical partners of the PcG recruiters. The PcG recruiters (shown in gray) interact with each other and with the proteins implicated in several functionally distinct regulatory activities: the PcG proteins, the proteins implicated in promoter pausing, the architectural proteins, Mediator, the SWI/SNF chromatin remodelers, and TFIID. The following interactions are shown between the PcG recruiters and different proteins: the solid blue lines—direct partners; the dotted black lines—interactions established by indirect methods. The interactions within the group of PcG recruiters: the solid orange lines—direct partners; the dotted green lines—interactions established by indirect methods.

**Figure 4 ijms-24-11394-f004:**
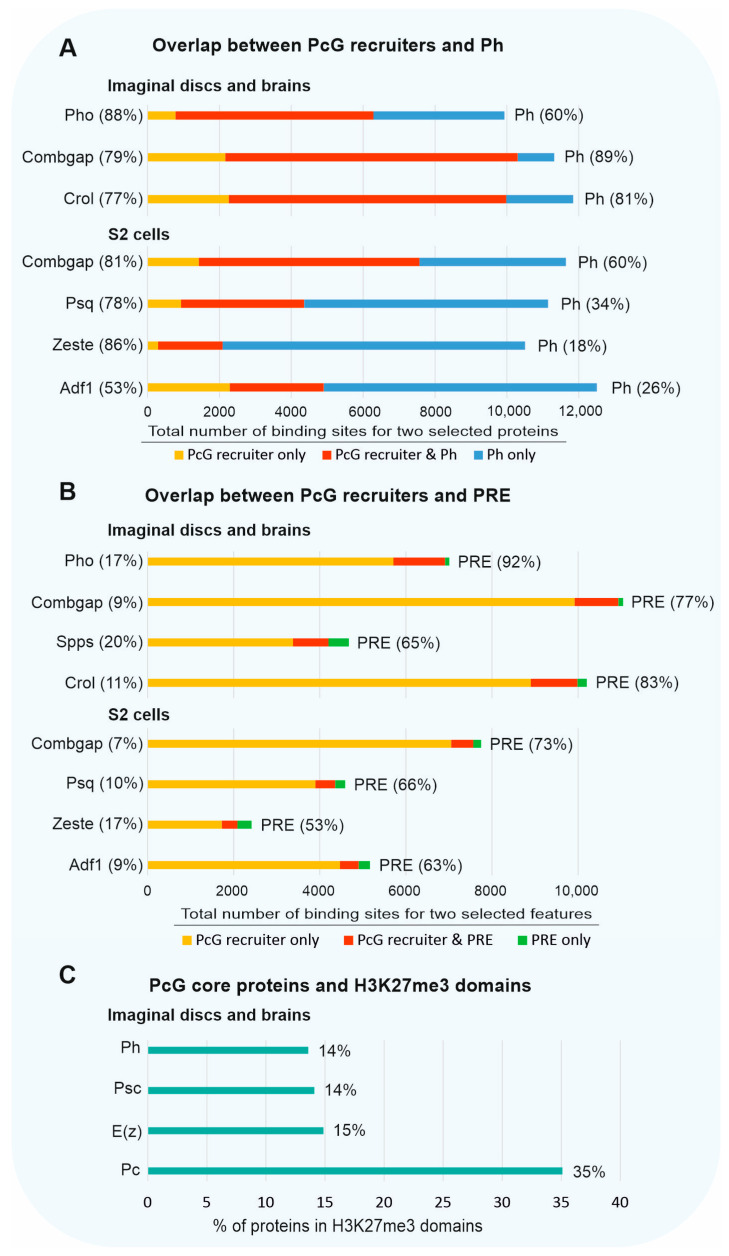
The genome-wide overlap between PcG recruiters and PcG proteins. (**A**) The percentages of overlap between PcG recruiters and Ph (PRC1) protein. The total number of binding peaks for two selected proteins are shown for each line. The PcG recruiter peaks that do not overlap with Ph are colored in yellow, the PcG recruiter peaks that do overlap with Ph are colored in red, the Ph peaks that are not co-bound by PcG recruiter are shown in blue. The percentages of overlapping peaks for each protein are indicated in the parentheses. (**B**) Overlap between PcG recruiters and PRE. The PcG recruiter peaks that do not overlap with PRE are colored in yellow, the PcG recruiter peaks that overlap with PRE are colored in red, the PRE peaks that are not co-bound by PcG recruiter are colored in green. (**C**) The percentage of the PcG core proteins’ binding sites located in the H3K27me3 domains.

**Figure 5 ijms-24-11394-f005:**
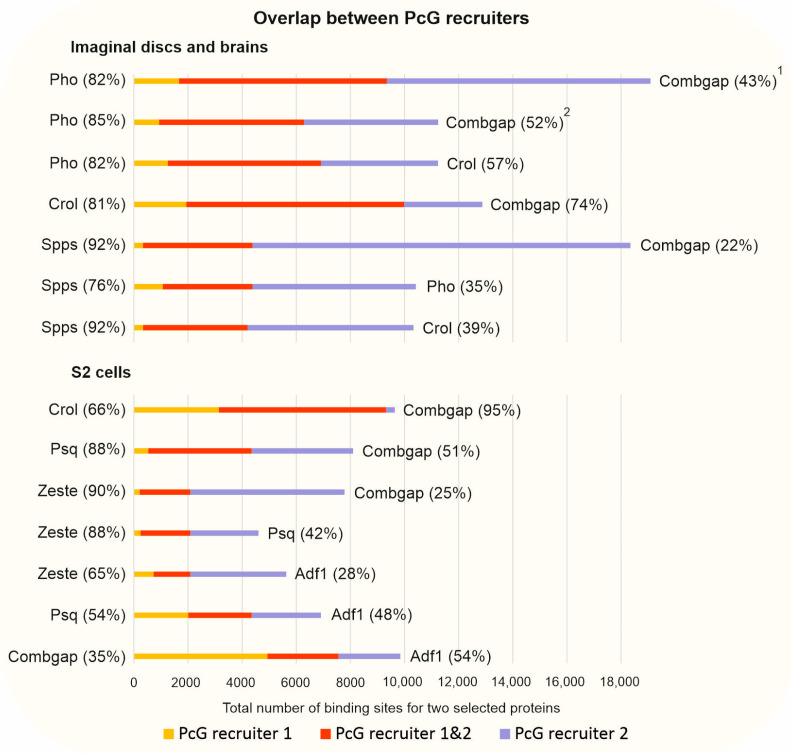
The genome-wide overlap between PcG recruiters. The total numbers of binding peaks for two selected PcG recruiters are shown for each line. The PcG recruiter 1 peaks (indicated on the left) that do not overlap with PcG recruiter 2 (indicated on the right) are colored in yellow, the PcG recruiters’ peaks that overlap with each other are colored in red, and the PcG recruiter 2 peaks that do not overlap with PcG recruiter 1 are colored in blue. For the Pho overlap with Combgap, the data derived from two different studies are presented: ^1^—[20]; ^2^—[59]. The other designations are as in Figure 4.

## Data Availability

All relevant data are within the paper.

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
