# Peer review of "Polycomb Recruiters Inside and Outside of the Repressed Domains"

_ijms, 2023, doi:10.3390/ijms241411394_

Round 1
Reviewer 1 Report
see the attached file.

The writing is fully understandable. Some moderate editing may make it more pleasant to read.
Author Response
>>Reply: We would like to thank the reviewer for a favorable review. Below, please, find our detailed point-by-point responses to the comments.
The manuscript by Erokhin et al gave a very comprehensive description of DNA binding proteins that interact with the Polycomb group repressive complexes (PRCs) and may function to target PRCs to specific genomic loci. The information presented is of considerable interest to the field.
Major concerns: 1.) Section 7 listed the overlaps between the genomic binding of recruiters and that of PRC and regions enriched for H3K27M3. The array of numbers is very hard on the reader. A few Venn diagrams would make this information accessible.
>>Reply: We have made an additional Figure (Figure 4) to illustrate the genome-wide overlaps between PcG recruiters and Ph, between PcG recruiters and PREs, and the positioning of the PcG core protein binding sites within, and by extension, outside of the H3K27me3 domains.
2.) Section 9 “Interactions between PcG recruiters and model of combinatorial recruitment” is also difficult to comprehend. Again, a diagram of the model is recommended.
>>Reply: We have made an additional Figure (Figure 5) to illustrate the genome-wide overlaps between different PcG recruiter proteins.
Minor concerns: 1.) The review is focused on enumerating the DNA binding proteins (recruiters) and their interaction with PcG. It did not dive into the details of the function of these proteins. So it is recommended to remove “Function of the” from the title, i.e. “ Polycomb recruiters inside and outside of the 2 repressed domains”.
>>Reply: We have changed the manuscript title to: “Polycomb recruiters inside and outside of the repressed domains”.
2.) References are of different formats. Some, like #15 and #61, had hyperlink to the doi but most of the others do not. It needs to be standardized. In general references with hyperlinks will make it easier for readers to follow.
>>Reply: We have added doi.org hyperlinks at the ends of all references that have doi (all except one).
3.) A few references were in languages other than English thus not accessible to this reviewer.
>>Reply: There are several articles originally published in Russian, however they all have English versions available. We have now added a doi.org hyperlink to each of the available English version. We thank reviewer for pointing out this oversight on our part.
Reviewer 2 Report
The review by Erokhyn et al describes recent studies in the area of transcriptional regulation. The authors revisit previous work on the epigenetic regulators PcG (Polycomb group) and TrxG (Trithorax group) that were initially described in Drosophila melanogaster. The PcG/TrxG proteins are recruited to cis-regulatory modules named PREs (Polycomb response elements) via a group of proteins named PcG recruiters. The review focus on many aspects of PcG recruiters in D. melanogaster, including their identification, role in transcription regulation, chromatin interaction and protein-protein interactions, which apparently have not been systematically addressed in the literature.
The only section that needs to be reviewed is number 7: “PcG recruiters are localized outside of H3K27me3 domains genome-wide”. The description of the overlap of PcG recruiters and PcG proteins is confusing (lines 284 to 296) and the authors should consider revising and/or creating a figure to illustrate the point.
Overall the review is well written, the theme of the review is well addressed and the conclusions that are drawn are coherent.
The quality of the English is good and only very minor adjustments are needed in the text.
Author Response
We would like to thank the reviewer for favorable review.
We have made additional Figure (Figure 4) to illustrate the data presented in the chapter 7:
- the genome-wide overlap between PcG recruiter proteins and Ph;
- the overlap between PcG recruiter proteins and PREs;
- the percentages of the PcG core proteins within the H3K27me3 domains.
Round 2
Reviewer 1 Report
The two new figures (4 and 5) are very informative. Nice addition to the review.